# Inhibition of 7-dehydrocholesterol reductase prevents hepatic ferroptosis under an active state of sterol synthesis

Naoya Yamada [1,2,3,9] ✉, Tadayoshi Karasawa [1,9] ✉, Junya Ito[4], Daisuke Yamamuro[5], Kazushi Morimoto [6], Toshitaka Nakamura [3], Takanori Komada [1], Chintogtokh Baatarjav [1], Yuma Saimoto[6], Yuka Jinnouchi[6], Kazuhisa Watanabe[7], Kouichi Miura[8], Naoya Yahagi [5], Kiyotaka Nakagawa [4], Takayoshi Matsumura [1,7], Ken-ichi Yamada [6], Shun Ishibashi[5], Naohiro Sata[2], Marcus Conrad [3] & Masafumi Takahashi [1] ✉

Recent evidence indicates ferroptosis is implicated in the pathophysiology of various liver diseases; however, the organ-specific regulation mechanism is poorly understood. Here, we demonstrate 7-dehydrocholesterol reductase (DHCR7), the terminal enzyme of cholesterol biosynthesis, as a regulator of ferroptosis in hepatocytes. Genetic and pharmacological inhibition (with AY9944) of DHCR7 suppress ferroptosis in human hepatocellular carcinoma Huh-7 cells. DHCR7 inhibition increases its substrate, 7-dehydrocholesterol (7-DHC). Furthermore, exogenous 7-DHC supplementation using hydroxypropyl β-cyclodextrin suppresses ferroptosis. A 7-DHC-derived oxysterol metabolite, 3β,5α-dihydroxycholest-7-en-6-one (DHCEO), is increased by the ferroptosis-inducer RSL-3 in *DHCR7*-deficient cells, suggesting that the ferroptosis-suppressive effect of DHCR7 inhibition is associated with the oxidation of 7-DHC. Electron spin resonance analysis reveals that 7-DHC functions as a radical trapping agent, thus protecting cells from ferroptosis. We further show that AY9944 inhibits hepatic ischemia-reperfusion injury, and genetic ablation of *Dhcr7* prevents acetaminophen-induced acute liver failure in mice. These findings provide new insights into the regulatory mechanism of liver ferroptosis and suggest a potential therapeutic option for ferroptosis-related liver diseases.

Iron-dependent cell death, ferroptosis, was originally identified as a new form of regulated cell death in RAS-mutated cancer cells. It occurs when intracellular glutathione peroxidase 4 (GPX4) is inhibited directly or indirectly by a decrease in glutathione (GSH) levels[1,2]. GPX4 inhibition accumulates iron-dependent lipid peroxides derived from polyunsaturated fatty acid (PUFA)-containing phospholipids, leading to cellular/subcellular membrane damage and eventually cell death without specific effector molecules. Although ferroptosis was initially discovered in cancer cells, recent research has revealed that ferroptosis is implicated in the pathogenesis of various human diseases. We and other investigators have demonstrated that ferroptosis plays a crucial role in the development of acute and chronic liver diseases, including acetaminophen (APAP)-induced acute liver failure, hepatic ischemia-reperfusion injury (IRI), hemochromatosis, non-alcoholic steatohepatitis (NASH), and alcoholic liver disease (ALD)[3–6]. However, the regulatory mechanism of ferroptosis in the liver remains less

understood. Furthermore, since the liver is a central organ for GSH synthesis and lipid and iron metabolism[6], which are the key factors to engaging ferroptosis, we postulated an unknown mechanism of ferroptosis regulation in the liver. Accordingly, we performed CRISPR/Cas9-mediated whole-genome screening and identified 7-dehydrocholesterol reductase (DHCR7), the terminal enzyme of the Kandutsch-Russell cholesterol pathway, as a regulator of ferroptosis in hepatocytes. Ferroptosis in human hepatocellular carcinoma-derived cells was suppressed by genetic and pharmacological inhibition of DHCR7, and also by exogenous supplementation with the substrate for DHCR7, 7-dehydrocholesterol (7-DHC). Furthermore, DHCR7 inhibition suppressed ferroptosis in murine primary hepatocytes in vitro and inhibited hepatic IRI in vivo. Our findings reveal a regulatory mechanism of hepatic ferroptosis and highlight DHCR7 and 7-DHC as potential therapeutic targets for ferroptosis-related liver diseases.

## Results

### DHCR7 inhibition suppresses ferroptosis in human hepatocellular carcinoma Huh-7 cells

To identify unknown regulators of ferroptosis in the liver, we performed CRISPR/Cas9-mediated whole-genome screening in human hepatocellular carcinoma Huh-7 cells. The cells were transduced with a lentiviral sgRNA library and treated with a lethal dose of the ferroptosis inducer RSL-3 in the presence or absence of linoleic acid (LA), which further promotes ferroptosis[7] (Fig. 1a). After selecting ferroptosis-resistant cells four times, we analyzed enriched sgRNAs in surviving cells and revealed that sgRNAs targeting *DHCR7* were markedly enriched in both RSL-3- and RSL-3/LA-resistant cells (34.2% of all sequenced gRNAs) (Fig. 1b and Supplementary Fig. 1a). To validate the screening, we generated *DHCR7*-ablated Huh-7 cells (polyclonal knockout [KO] cells) and confirmed defective enzymatic activities (Supplementary Fig. 1b–f). Similar to a screening experiment, *DHCR7*-ablated cells were resistant to RSL-3-induced ferroptosis (Fig. 1c–e). To exclude the possibility that the effect was limited to RSL-3-induced ferroptosis, we examined other types of ferroptosis. Ferroptosis induced by artesunate or cystine deprivation was also suppressed in *DHCR7*-ablated cells (Fig. 1f, g). The inhibitory effect of *DHCR7* ablation on RSL-3-induced ferroptosis was also confirmed in Huh-7 cells harboring a large deletion around exon 4 of *DHCR7* (Supplementary Fig. 2a–e).

To further confirm the genetic proof of ferroptosis regulation by DHCR7, we generated single clonal *Dhcr7* knockout (KO) cells (C.344dupA:p.Y114*) in 4-hydroxytamoxifen-induced *Gpx4*-knockout mouse embryonic fibroblasts (Pfa1 cells) (Supplementary Fig. 3a and b)[8]. Consistent with Huh-7 cells, RSL-3-induced ferroptosis was suppressed in *Dhcr7* KO Pfa1 cells (Supplementary Fig. 3c). Moreover, the ferroptosis suppressive effect was diminished by the induction of human *DHCR7* in *Dhcr7* KO Pfa1 cells, while not by mutant *DHCR7* which lacks enzymatic activity (R228Q, T93M) (Supplementary Fig. 3d).

To validate the suppression of ferroptosis by DHCR7 inhibition, we used the DHCR7 inhibitor AY9944 and confirmed that AY9944 suppressed RSL-3-induced ferroptosis in Huh-7 cells in a dose-dependent manner (Fig. 1h, i). Furthermore, we generated *GPX4*-ablated cells which were maintained with a minimum dose of Fer-1 and underwent ferroptosis at 48 h after Fer-1 withdrawal (termed "spontaneous ferroptosis"). AY9944 suppressed this spontaneous ferroptosis (Fig. 1j).

### DHCR7 inhibition prevents lipid peroxidation

Lipid peroxidation determined by C11-BODIPY[581/591] staining, a hallmark of ferroptosis, was suppressed in *DHCR7*-ablated cells and by AY9944 treatment (Fig. 2a, b, Supplementary Fig. 4a). Furthermore, LC-MS/MS analysis revealed that various kinds of lipid peroxidation products that are known to be characteristic of ferroptosis were increased by

RSL-3 treatment in Huh-7 cells, and these increases were apparently decreased in *DHCR7*-ablated cells (Fig. 2c). The same trend was observed in AY9944-treated Huh-7 cells (Fig. 2d), indicating that lipid peroxidation related to ferroptosis is suppressed by DHCR7 inhibition. The expression of major ferroptosis-regulating molecules such as SLC7A11/xCT, GPX4, ACSL4, FSP1(*AIFM2*), and DHODH, or the expression of iron metabolism proteins (DMT1, FTH, FTL, and SLC40A1 [Ferroportin]) was not influenced by *DHCR7* ablation (Supplementary Fig. 4b–d), suggesting that DHCR7 is a ferroptosis regulator.

### 7-DHC accumulation, but not cholesterol deprivation, prevents ferroptosis

DHCR7 is an enzyme that acts in the final step of the cholesterol biosynthesis pathway and converts 7-DHC possessing conjugated diene to cholesterol[9]. Therefore, DHCR7 inhibition leads to intracellular accumulation of 7-DHC[10]. Indeed, we measured the intracellular 7-DHC levels and found that they were dramatically increased in *DHCR7*-ablated cells or AY9944-treated cells, although they were not detected in untreated cells (Fig. 3a, b). To determine whether accumulated 7-DHC results in ferroptosis-resistance in *DHCR7*-ablated cells, we transduced Huh-7 cells with sgRNA targeting *DHCR7* and sterol-C5-desaturase (*SC5D*), which introduces a C5-6 double bond into lathosterol (Fig. 3c and Supplementary Fig. 4e)[11]. Expectedly, the 7-DHC accumulation induced by *DHCR7* ablation was markedly decreased by transduction of gRNA targeting *SC5D* (Fig. 3d). Furthermore, RSL-3 or cystine deprivation-induced ferroptosis suppression in *DHCR7*-ablated cells was canceled by ablation of *SC5D* (Fig. 3e, f).

Next, we investigated the possibility that decreased cholesterol levels and altered lipid composition by DHCR7 inhibition affect ferroptosis sensitivity. However, cholesterol depletion by the cholesterol-extracting agent Methyl-β-cyclodextrin (Mβ-CD) failed to suppress ferroptosis (Fig. 3g). Furthermore, cellular levels of cholesterol and phosphatidyl ethanolamine (PE) were not decreased in *DHCR7*-ablated cells with a slightly increased expression of genes involved in cholesterol synthesis (Supplementary Fig. 5a–e). Consistent with our results, a previous study demonstrated that a minimal effect of *DHCR7* knockout on cholesterol levels was observed in liver-specific *Dhcr7*-KO (*Dhcr7*-LKO) mice[12]. Using RNA-seq datasets of *Dhcr7*-LKO mice, we revealed that glycerophospholipid metabolism was unaltered in *Dhcr7*-LKO mice, whereas several enzymes involved in cholesterol synthesis (*Hmgcs1*, *Fdps*, and *Cyp51*) were upregulated in *Dhcr7*-LKO mice (Supplementary Fig. 6). In contrast, although a modest decrease in cellular cholesterol was induced by treatment with AY9944, cellular PC and PE levels were unchanged by AY9944 (Supplementary Fig. 5f–j). These findings suggest that the accumulation of 7-DHC, but not cholesterol deprivation, is the main mechanism by which DHCR7 inhibition suppresses ferroptosis.

### 7-DHC works as a radical trapping agent and is converted to oxysterol metabolites

To investigate whether the accumulated 7-DHC suppresses ferroptosis, we evaluated the effect of exogenous 7-DHC supplementation using a water-soluble 7-DHC-HPβCD inclusion complex. The intracellular 7-DHC levels were successfully increased when the 7-DHC-HPβCD inclusion complex was added (Fig. 4a). In this setting, exogenous 7-DHC supplementation suppressed both RSL-3-induced and spontaneous ferroptosis (Fig. 4b, c).

7-DHC is known for its highly oxidizable property and its modification to various kinds of oxysterol metabolites[13]. In particular, 7-DHC has been reported to easily react with peroxy radicals, compared to arachidonic acid, the oxidation of which is critical for ferroptosis[14]. Notably, the elevation of 7-DHC levels in AY9944-treated cells was diminished by RSL-3 (Fig. 3b); therefore, we assumed that 7-DHC could be metabolized to other forms in the process of suppressing ferroptosis. We hypothesized that 7-DHC is oxidized and

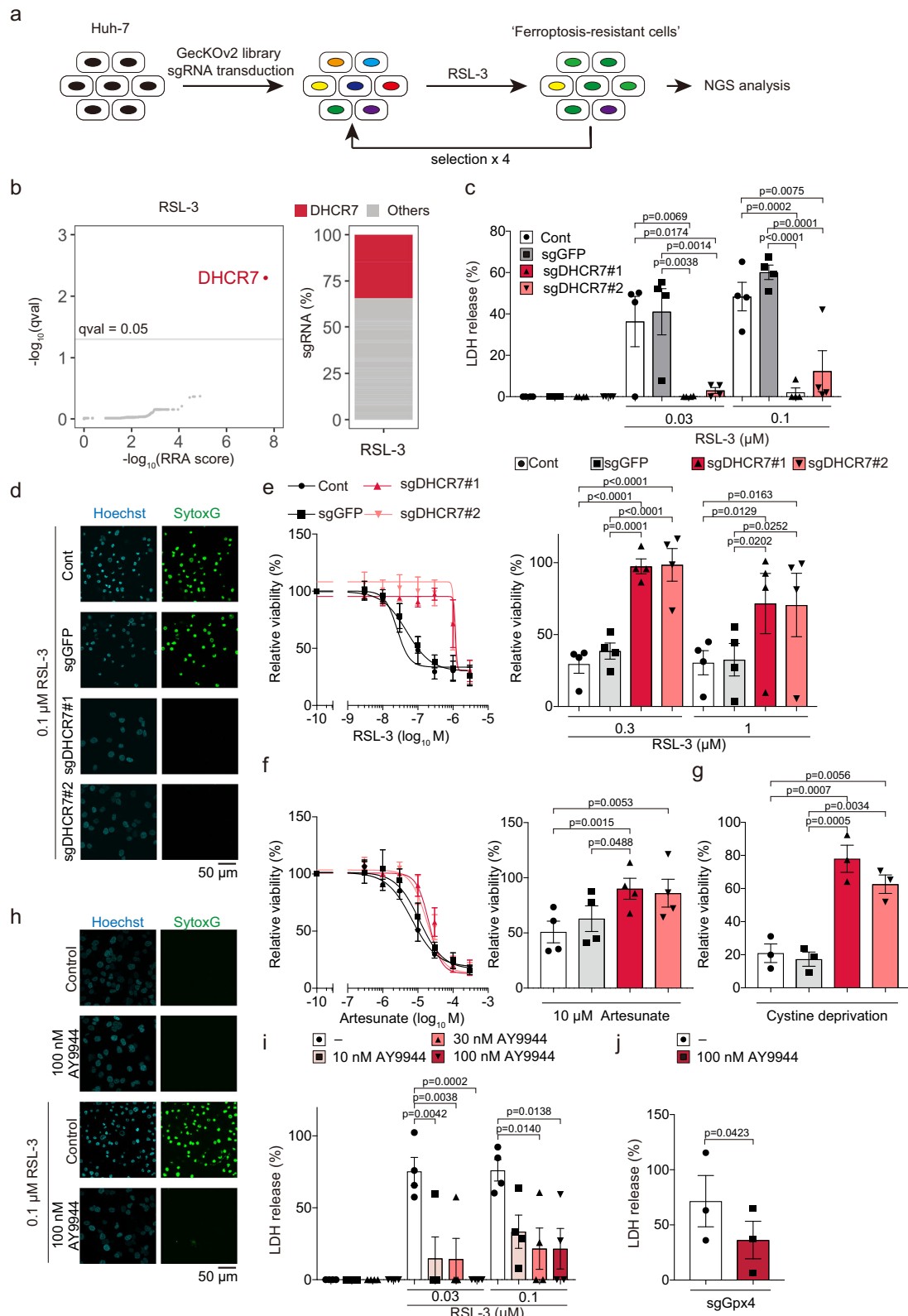

functions as a radical scavenger to prevent ferroptosis[13,15]. To test this hypothesis, we analyzed the radical trapping activity of 7-DHC by using ESR and NBD-Pen[16]. 7-DHC attenuated the reduction of TMPOL ESR signal caused by the reaction with arachidonic acid radicals (Fig. 4d and Supplementary Fig. 7a). Similarly, 7-DHC suppressed the increase in fluorescence intensity of NBD-PEN, a probe that detects lipid radicals (Fig. 4e), suggesting that 7-DHC can act as a radical trapping agent.

Subsequently, we performed LC-MS/MS analysis to assess DHCEO, one of the major 7-DHC-derived oxysterol metabolites[13,15], in RSL-3-treated and untreated *DHCR7*-ablated cells (Supplementary Fig. 7b). As expected, DHCEO was detected only in *DHCR7*-ablated cells, and was significantly increased by RSL-3 (Fig. 4f and g). DHCEO was also increased in AY9944-treated cells, and its increase was enhanced by RSL-3 treatment (Supplementary Fig. 7c and d), suggesting that 7-DHC

**Fig. 1 | DHCR7 inhibition suppresses ferroptosis in Huh-7 cells. a** Strategy of CRISPR/Cas9-mediated whole-genome screening. Huh-7 cells were treated with RSL-3 (0.1 μM) for 24 h. The ferroptosis-resistant cells were selected 4 times, and the genomic DNA was purified and analyzed by NGS. (**b**) sgRNAs targeting *DHCR7* were enriched in ferroptosis-resistant cells. (**c, d**) Control (sgGFP) and *DHCR7*-ablated (sgDHCR7) Huh-7 cells were treated with or without RSL-3 (0.03 and 0.1 μM) for 24 h. Cytotoxicity and cell death were assessed by an LDH release assay and SYTOX Green, respectively. Control (sgGFP) and *DHCR7*-ablated (sgDHCR7) Huh-7 cells were treated with or without (**e**) RSL-3 or (**f**) artesunate for 24 h, and (**g**) cysteine deprivation for 18 h, respectively. Cell viability was assessed by the MTT assay.

**h, i** Huh-7 cells were treated with RSL3 for 24 h with or without AY9944 1 h prior to RSL-3 administration. Cytotoxicity and cell death were assessed by an LDH release assay and SYTOX Green, respectively. (**j**) Control (sgGFP) and *GPX4*-ablated (sgGPX4) cells were maintained with Fer-1 (0.1 μM), and then pretreated with Fer-1 (0.5 μM) or AY9944 (100 nM) for 1 h, followed by Fer-1 withdrawal. Cytotoxicity at 48 h was assessed by an LDH release assay. (**d h**) Data are representative of three independent experiments (**c, e**–**g i, j**). Data are means of (**g and j**) three or (**c, e, f, and i**) four independent experiments and expressed as dot plots and means ± SEM. Statistical significance was calculated using two-way ANOVA with Tukey's post hoc test or Student's t-test.

is primarily oxidized instead of membrane phospholipids and suppresses ferroptosis.

## The pharmacological effect of AY9944 on ferroptosis inhibition is due to DHCR7 inhibition

AY9944 has been identified as a DHCR7 inhibitor, but is also known to target various molecules through off-target effects[17]. In particular, AY9944 inhibits DHCR14a (encoded by *TM7SF2*), DHCR14b (*LBR*), or sterolΔ8-Δ7 isomerase (*EBP*), which are upstream of DHCR7 in the cholesterol synthesis pathway (Supplementary Fig. 8a). To exclude the possibility that these enzymes are involved in the suppression of ferroptosis by AY9944, we evaluated accumulation of their substrates, 14-dehydro zymostenol (14DZyme) and zymostenol (Zyme). As reported previously[18], low dose AY9944 treatment (10–100 nM) selectively increased 7-DHC, whereas high dose AY9944 treatment (300–1000 nM) also increased 14DZyme, a substrate for DHCR14 (Supplementary Fig. 8b and c). To further validate the involvement of these enzymes, we generated *TM7SF2*-, *LBR*-, and *EBP*-ablated cells (Supplementary Fig. 9a and b) and treated the cells with RSL-3 in the presence or absence of AY9944. Unlike *DHCR7*-ablated cells, ferroptosis was not suppressed in these ablated cells (Supplementary Fig. 9c), indicating that inhibition of these enzymes was not the cause of AY9944-mediated ferroptosis suppression. Consistent with the finding in *SC5D*-ablated cells, the ferroptosis-suppressive effect of AY9944 was diminished in *LBR*- and *EBP*-ablated cells. In addition, we tested BM15766, another DHCR7 inhibitor, and found that BM15766 also suppressed ferroptosis (Supplementary Fig. 8d). These results suggest that the suppression of ferroptosis by AY9944 was not due to its off-target effects via other enzymes.

## Effect of DHCR7 inhibition is dependent on sterol synthesis activity

Next, we studied the effects of 7-DHC and AY9944 on cells from various organs. Exogenous 7-DHC supplementation almost completely suppressed ferroptosis in ferroptosis-sensitive cancer cells: human fibrosarcoma HT1080 and human ovarian clear cell carcinoma OVISE cells (Fig. 5a). Unexpectedly, however, AY9944 had little effect on ferroptosis in these cells (Fig. 5b). We also examined several human hepatocellular carcinoma-derived cells, and found that AY9944 had no effects on RSL-3-induced ferroptosis in these cells (Supplementary Fig. 10a). These findings prompted us to analyze the expression levels of *DHCR7* in various cancer cells. A gene expression database of cancer cell lines (CellExpress, http://cellexpress.cgm.ntu.edu.tw) revealed that among various kinds of cancer cell lines, *DHCR7* expression in liver cancer cells was higher than that in most cancer cells of other origins (Supplementary Fig. 10b)[19]. Indeed, we confirmed that the *DHCR7* expression in Huh-7 cells is much higher than that in other ferroptosis-sensitive cancer cells, including liver cancer (Supplementary Fig. 10c). Since enzymes involved in cholesterol synthesis are coordinately regulated by sterol regulatory element binding proteins (SREBPs)[20], we compared the expression of *DHCR7* and its upstream enzymes in cholesterol synthesis across multiple cancer cells registered in the Cancer Cell Line Encyclopedia (CCLE). The expression of *HMGCR* and *SC5D* was weakly correlated with *DHCR7* expression (Fig. 5c).

Furthermore, Huh-7 cells exhibit a considerably higher expression of these enzymes than other cell lines. To estimate the activity of cholesterol synthesis among these cell lines, we scored the expression of genes listed in the Cholesterol biosynthesis gene set in the Reactome Pathway Database. Cholesterol synthesis in hepatocellular carcinoma was more active than that in other types of cells (Fig. 5d). Among hepatocellular carcinoma, Huh-7 cells exhibit the highest expression levels of cholesterol synthesis genes. Thus, we hypothesized that the suppression of ferroptosis by DHCR7 inhibition is effective in cells with active cholesterol synthesis. The expression of the cholesterol biosynthesis gene set in AY9944-insensitive HT1080 and OVISE cells is lower than that in Huh-7 cells (Fig. 5e). We further assessed whether induction of cholesterol synthesis by SREBP-1 sensitizes AY9944-insensitive HT1080 cells. Transduction of the nuclear form of human SREBP-1a induced the expression of genes involved in cholesterol synthesis, including *DHCR7*, *SC5D*, and *HMGCR* in HT1080 cells (Fig. 5f and Supplementary Fig. 10d). Indeed, AY9944 inhibited RSL-3-induced ferroptosis under the induction of SREBP-1 (Fig. 5g, h). These results suggest that the DHCR7 inhibitor prevents ferroptosis in the cells actively synthesizing sterol, whereas 7-DHC prevents ferroptosis in a cell type-independent fashion.

## AY9944 prevents ferroptosis of normal hepatocytes in vitro and in vivo

Consistent with its fundamental function as the key enzyme in the synthesis of cholesterol, *DHCR7* expression in the liver is particularly high in normal tissues and organs (RefEx, https://refex.dbcls.jp)[21] (Fig. 6a). Therefore, we investigated the role of DHCR7 in primary hepatocytes. AY9944 suppressed ferroptosis induced by RSL-3, iron overload, and cysteine deprivation in murine primary hepatocytes (Supplementary Fig. 11a–c). We have recently shown that ferroptosis contributes to the development of hepatic IRI[4]. Intraperitoneal injection of AY9944 significantly attenuated liver injury, as evaluated by serum levels of AST and ALT, after hepatic IRI in male mice (Figs. 6b and 6c, Supplementary Fig. 11d, e). Notably, several lipid peroxidation products of PC and PE, characteristic of ferroptosis, were increased by IRI, which was decreased by AY9944 treatment (Fig. 6d and Supplementary Fig. 11f). 7-DHC levels in the liver tissue and serum were increased by AY9944, which was decreased by IRI (Fig. 6e). Although DHCEO was detected only in AY9944-treated mice and could not be quantified because of its very low level (Supplementary Fig. 11g, h), these findings are consistent with cell experiments. Because pharmacological inhibition of cholesterol synthesis may cause adverse effects, we assessed the effects of AY9944 on cholesterol levels, cell proliferation, and steroid hormones. In this setting, AY9944 had no effects on liver free cholesterol, liver weight, or PC, or PE levels (Supplementary Fig. 12a). AY9944 also did not affect cell proliferation, as determined by BrdU incorporation in the small intestines and white blood cells (Supplementary Fig. 12b–d). The levels of cortisol, but not aldosterone, were increased in the AY9944-treated group (Supplementary Fig. 12e).

To further evaluate the suppressive effects of DHCR7 inhibition on hepatic ferroptosis in vivo, we examined whether genetic ablation of *Dhcr7* could prevent APAP-induced acute liver failure, another mouse

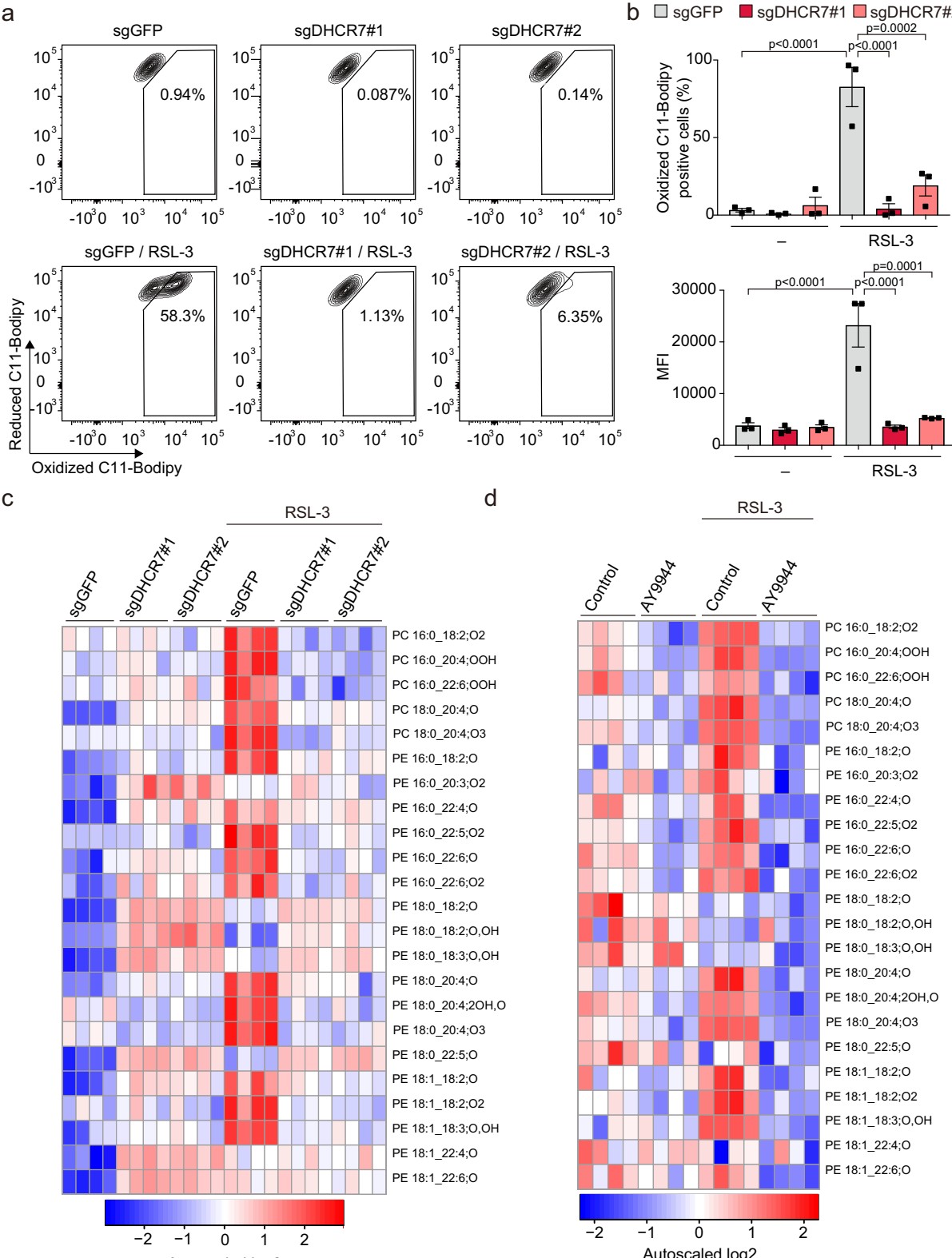

**Fig. 2 | DHCR7 inhibition suppresses lipid peroxidation.** Control (sgGFP) and *DHCR7*-ablated (sgDHCR7) Huh-7 cells were treated with RSL-3 (0.1 μM) for 6 h. The fluorescence intensity of C11-BODIPY[581/591] was analyzed by flow cytometry. **a** Representative plot of C11-BODIPY[581/591]-stained cells. **b** The number of oxidized C11-BODIPY[581/591]-positive cells and the mean fluorescence intensity of oxidized C11-BODIPY[581/591] were analyzed. (**c, d**) Control (sgGFP) and *DHCR7*-ablated (sgDHCR7)

Huh-7 cells or A9944-treated cells were treated with RSL3 (0.1 μM) for 16 h, and lipid peroxides were measured by LC-MS/MS. **a** Data are representative and **b** means of three independent experiments and expressed as dot plots and means ± SEM. **c, d** Data are from four replicates. Statistical significance was calculated using two-way ANOVA with Tukey's post hoc test.

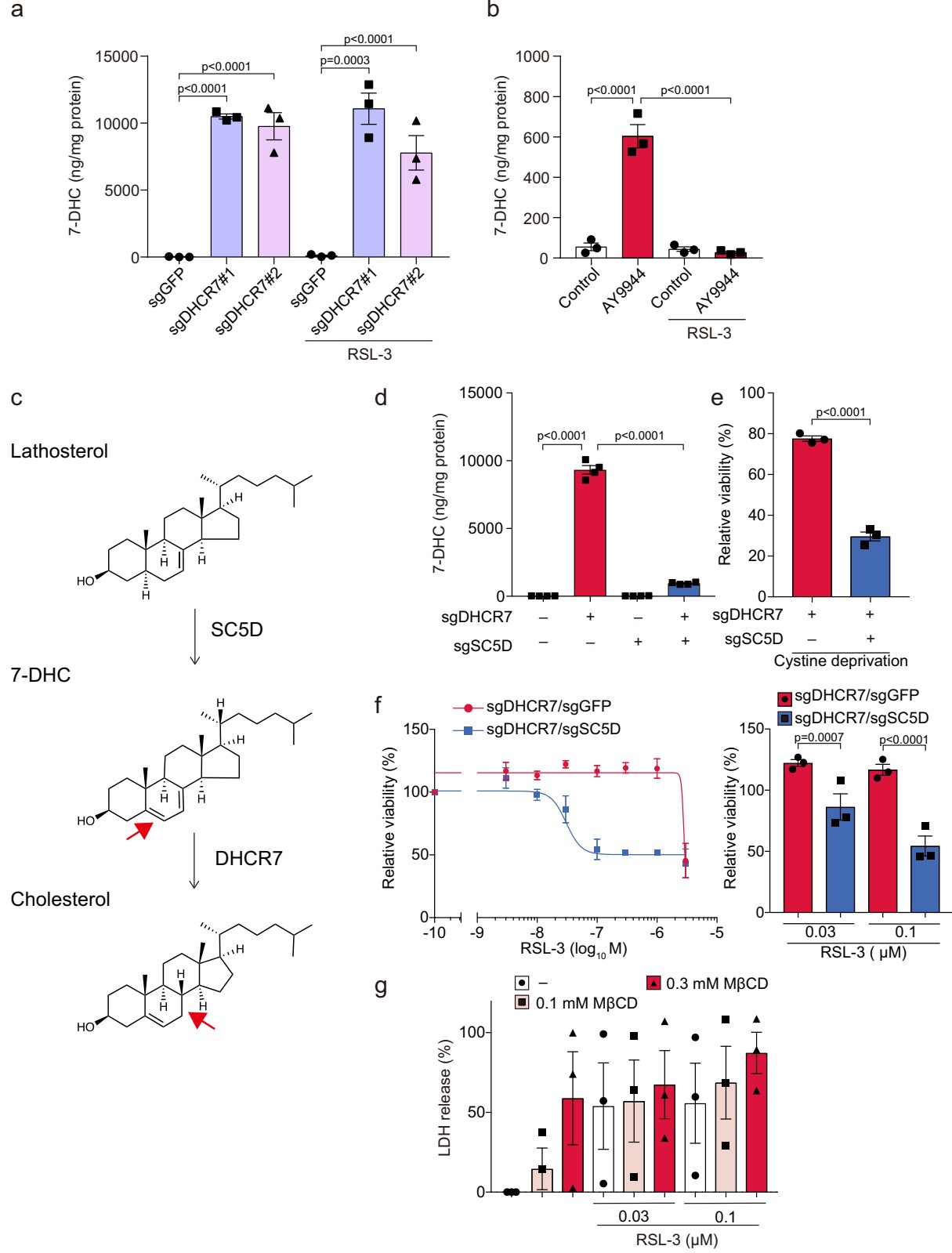

model of ferroptosis-related liver injury accompanied by lipid peroxidation[3,22]. We applied hydrodynamic injection in male mice which enables gene transfer to zone 3 in the liver to introduce plasmid encoding a pair of sgRNA targeting *Dhcr7* and SpCas9[23]. Although the efficiency of gene ablation was not sufficient to decrease the net expression of *Dhcr7* in the liver, gene ablation and expression of truncated *Dhcr7* were confirmed (Supplementary Fig. 13). Moreover, partial

ablation of the *Dhcr7* gene attenuated APAP-induced liver injury (Fig. 6f, g). These findings suggest that DHCR7 is a potential therapeutic target for treating or preventing liver diseases related to ferroptosis.

## Discussion

Ferroptosis is involved in the pathogenesis of many liver diseases and is regarded as a potential target for these disorders; however, the

**Fig. 3 | Accumulation of 7-DHC by DHCR7 inhibition, but not cholesterol deprivation, suppresses ferroptosis. a** Control (sgGFP) and *DHCR7*-ablated (sgDHCR7) Huh-7 cells were treated with or without RSL-3 (0.1 μM) for 16 h. **b** Huh-7 cells were pretreated with AY9944 (100 nM) for 1 h, and then treated with RSL-3 for 16 h. **a, b** Intracellular 7-DHC levels were assessed using LC-MS/MS analysis. *n* = 3, respectively. **c** Schematic diagram of a cholesterol synthesis pathway via lathosterol. The introduced or reduced double bond is shown by a red arrow. **d** Huh-7 cells transduced with sgDHCR7 and sgSC5D. Intracellular 7-DHC levels were assessed using LC-MS/MS analysis. Huh-7 cells transduced with sgDHCR7 and sgSC5D were treated with (**e**) cysteine deprivation for 18 h or (**f**) RSL-3 for 24 h. Cell viability was assessed by the MTT assay. **g** Cells were pretreated with MβCD (0.1 mM) for 1 h, and then treated with RSL-3 for 24 h. Cytotoxicity was assessed by an LDH release assay. (**a, b**, and **d**) Data are from (**a** and **b**) three or (**d**) four replicates. **e–g** Data are means of three independent experiments. Statistical significance was calculated using two-way ANOVA with Tukey's post hoc test. Data are expressed as dot plots and means ± SEM.

regulatory mechanism of liver ferroptosis is poorly understood[5]. In the present study, using the whole-genome screening approach, we identified DHCR7 as a regulator of ferroptosis in hepatocytes. DHCR7 inhibition or exogenous supplementation with DHCR7's substrate 7-DHC suppressed ferroptosis in Huh-7 cells and normal primary hepatocytes. Furthermore, DHCR7 inhibition ameliorated hepatic IRI and APAP-induced liver injury in vivo. These findings demonstrate that targeting DHCR7 has the potential for treating and preventing ferroptosis-related diseases.

Unlike other sterol intermediates in the Kandutsch-Russell pathway, 7-DHC possesses conjugated dienes that characterize its reactive properties. For instance, 7-DHC is metabolized to vitamin D by ultraviolet light B in the skin and plays an important role in calcium homeostasis and immune function[9]. Previous reports also showed that 7-DHC could be extremely prone to free radical oxidation. In particular, Porter et al.[14] reported that the calculated *k*p value of 7-DHC to react with peroxy radicals is about 11 times higher than that of arachidonic acid, oxidation of which is absolutely key for initiating ferroptosis. Multiple studies have implied that 7-DHC and its oxidation products are toxic[24,25]; however, the current study demonstrated unique aspects of 7-DHC as a radical trapping agent, which suppresses ferroptosis. Indeed, we observed that DHCR7 inhibition increased intracellular levels of 7-DHC and its major oxysterol metabolite, DHCEO, which was further increased by RSL-3 treatment. We assume that 7-DHC is rapidly oxidized prior to membrane phospholipid peroxidization, thereby preventing ferroptosis. However, more than 10 kinds of 7-DHC-derived oxysterol metabolites have been identified so far[13]. Thus, further studies to assess the amounts of each metabolite will be needed to clarify the mechanism through which oxysterol regulates ferroptosis and how cells escape from oxidative damage by these oxysterols.

In the present study, we found two types of ferroptosis-inhibiting strategies: exogenous 7-DHC delivered as an inclusion complex with HPβCD and DHCR7 inhibition. Whereas the suppressive effect of DHCR7 inhibition was limited to cells actively synthesizing sterol, exogenous 7-DHC suppressed ferroptosis in a cell type-independent fashion. Although both endogenously synthesized and exogenously delivered 7-DHC suppressed ferroptosis, the exogenous 7-DHC treatment required higher 7-DHC levels to suppress ferroptosis than that under DHCR7 inhibition. Although the mechanism of this discrepancy is unclear, the difference in the intracellular localization of exogenous and endogenous 7-DHC might affect the efficiency of ferroptosis suppression. Similar to cholesterol, 7-DHC is synthesized in the endoplasmic reticulum and delivered to lipid rafts of plasma membrane[26]. In contrast, the inclusion complex directly delivers sterol to the plasma membrane[27]. Further investigations will be needed to determine the intracellular localization of 7-DHC that can suppress ferroptosis.

A comparative analysis of cancer cells revealed that sensitivity to DHCR7 inhibition was different among the cell lines, probably because of the difference in 7-DHC accumulation. The effect of DHCR7 inhibition was most profound in Huh-7 cells, which highly express enzymes involved in cholesterol synthesis regulated by SREBPs, indicating that active sterol synthesis is required for ferroptosis-resistance induced by DHCR7 inhibition. Concordantly, insensitivity to RSL-3-induced ferroptosis in *DHCR7*-ablated cells was cancelled by the ablation of *SC5D*,

the essential enzyme for 7-DHC production. Previous studies have suggested that SREBP-mediated cholesterol synthesis is active in proliferative cancer cells[28,29]. Therefore, DHCR7 inhibition might alter ferroptosis sensitivity in proliferative cancer cells. On the other hand, the major site of sterol synthesis among tissues is the liver, where 7-DHC is actively produced[30]. Our data clearly suggest that pharmacological inhibition of DHCR7 effectively increased hepatic 7-DHC levels and prevented ferroptosis-associated liver injury. Kanuri et al. reported that hepatocyte-specific *Dhcr7* knockout exhibits elevated 7-DHC levels but exhibits no apparent signature of Smith-Lemli-Opitz syndrome (SLOS). Therefore, we assume that hepatic inhibition of DHCR7 could be a potential target of ferroptosis-associated liver injury.

Considering the clinical application of ferroptosis regulation, the inhibition of ferroptosis is expected to be a therapeutic option for ferroptosis-related liver diseases, such as hepatic IRI, hemochromatosis, NASH, and APAP-induced acute liver failure[5,10]. On the other hand, the induction of ferroptosis is also expected to be a novel therapeutic strategy for cancer. Ferroptosis was originally found in a project searching for new anticancer drugs, and conventional chemotherapy-resistant cancer cells are known to be sensitive to ferroptosis[1,31,32]. The induction of ferroptosis in cancer cells is expected to be an innovative therapy. On the other hand, chronic liver disease progresses to liver fibrosis and cirrhosis, leading to liver cancer. Liver fibrosis is driven by activated hepatic stellate cells, which have recently been reported to be highly susceptible to ferroptosis. Therefore, induction of ferroptosis in activated hepatic stellate cells could be a new therapeutic approach for liver fibrosis. The biggest concern with this ferroptosis induction strategy is adverse side effects on ferroptosis-susceptible tissues/organs, including the liver. Considering the tissue-specific expression patterns of DHCR7, its inhibition can be a more specific approach for suppressing ferroptosis in hepatocytes. Moreover, it seems likely that DHCR7 inhibition can reduce the side effects on the liver in ferroptosis-inducing therapy against progressive liver fibrosis or chemotherapy-resistant cancers.

In addition, mutations of the DHCR7 gene in humans cause SLOS, which is characterized by multiple congenital anomalies, including microcephaly, developmental delay, typical facial appearance, and cardiac abnormalities[33,34]. The patients show low serum cholesterol levels, and increased serum and tissue 7-DHC levels[35]. Therefore, an adverse effect of DHCR7 inhibition must be considered for its clinical application. Indeed, AY9944 treatment in a pregnant rat is used for a model of SLOS[15]. Since decreased cholesterol synthesis is regarded as a major cause of teratogenic activity in an animal model of SLOS[36], we evaluated the cholesterol levels in *DHCR7*-ablated or AY9944-treated conditions. While the accumulation of 7-DHC was detected, the cellular cholesterol levels were unchanged in *DHCR7*-ablated Huh-7 cells. Furthermore, a single injection of AY9944 did not decrease hepatic cholesterol levels in mice. Similarly, a previous report suggested that cholesterol levels are unchanged in liver-specific *Dhcr7* KO mice that exhibit no apparent SLOS signature[12]. Therefore, we postulate that cholesterol derived from lipoprotein can compensate for impaired de novo cholesterol synthesis in the liver. On the other hand, the accumulated 7-DHC and 7-DHC-derived oxysterol metabolites, including DHCEO might contribute to SLOS phenotype[15]. Hence, although no serious impairment was observed in our single-dose experiments

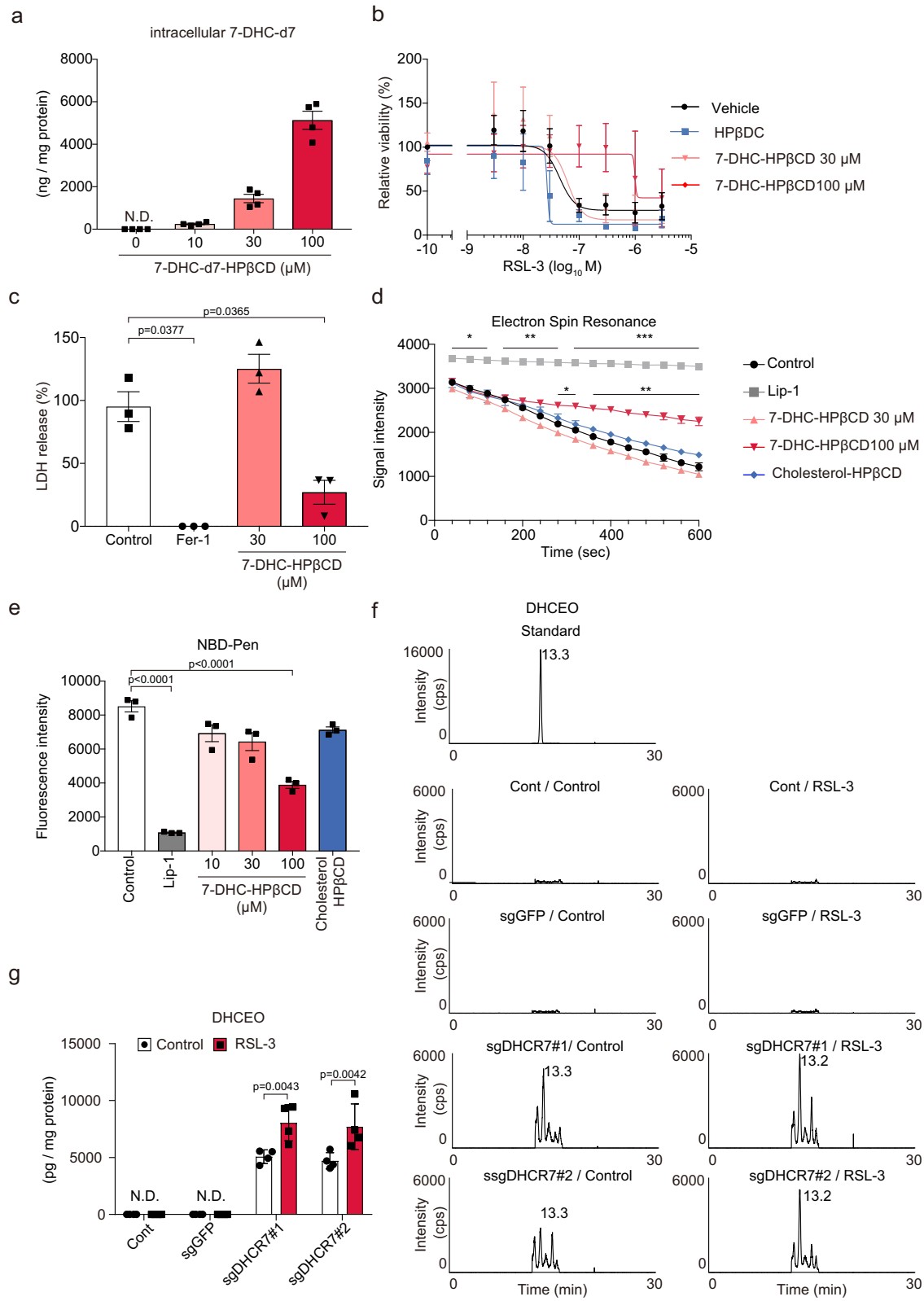

against an acute liver injury model, we cannot rule out the possibility that long-term DHCR7 inhibition can have adverse effects. Thus, further investigations are necessary for the development of safe clinical applications to protect against hepatic ferroptosis by DHCR7 inhibition or 7-DHC supplementation.

In conclusion, our results suggest that the terminal enzyme of cholesterol biosynthesis (DHCR7) and its substrate (7-DHC) suppress hepatic ferroptosis and provide new insights into the mechanism of liver ferroptosis.

## Methods

### Cell culture and reagents

Huh-7 (human hepatocellular carcinoma), OVISE (human ovarian clear cell carcinoma), and HT1080 (human fibrosarcoma) cells were

**Fig. 4 | 7-DHC acts as a radical trapping agent to prevent ferroptosis. a** Huh-7 cells were treated with a 7-DHC-d7-HPβCD inclusion complex for 16 h, and the intracellular 7-DHC-d7 levels were measured by LC-MS/MS. **b** Huh-7 cells were pretreated with 7-DHC-HPβCD for 1 h and then treated with RSL-3 for 24 h. Cell viability was assessed by the MTT assay. **c** GPX4-ablated cells were pretreated with 7-DHC-HPβCD for 1 h, followed by Fer-1 withdrawal. Cytotoxicity at 48 h was assessed by an LDH release assay. **d** ESR signal intensity of TEMPOL (50 μM) after the addition of LOX (15 μg/ml) and AA (250 μM) with 7-DHC-HPβCD (30 μM or 100 μM), cholesterol-HPβCD (100 μM) or Lip-1 (100 μM). **e** Fluorescence intensities of NBD-Pen (5 μM) 5 min after the addition of LOX (5 μg/ml) and AA (500 μM) with 7-DHC-HPβCD (10 μM, 30 μM or 100 μM), cholesterol-HPβCD (100 μM) or Lip-1 (100 μM). **f, g** Control (sgGFP) and DHCR7-ablated (sgDHCR7) Huh-7 cells were treated with or without RSL-3 (0.1 μM) for 16 h. Intracellular DHCEO levels were assessed using LC-MS-MS analysis. Data are from (**d** and **e**) three or (**a** and **g**) four replicates. **b, c**) Data are means of three independent experiments. Statistical significance was calculated using one-way or two-way ANOVA with Tukey's post hoc test. Data are expressed as dot plots and means ± SEM. *$p < 0.05$, **$p < 0.01$.

obtained from the Japanese Collection of Research Bioresources (JCRB) Cell Bank (Japan). PLC/PRF/5 and HLE (human hepatocellular carcinoma) cells were obtained from The Health Science Research Resource Bank (HSRRB, Japan). SKHep1 (human hepatocellular carcinoma) cells were obtained from ATCC. 4-hydroxytamoxifen-induced Gpx4-knockout mouse embryonic fibroblasts (Pfa1 cells) was kindly provided by Dr. Marcus Conrad. The cells were cultured in Dulbecco's modified Eagle's medium (DMEM; Wako, Osaka, Japan) supplemented with 10% fetal calf serum (FCS; Dainippon Pharmaceutical Company, Osaka, Japan) and antibiotics. LentiX293T (Takara Bio, Shiga, Japan) cells were cultured in DMEM supplemented with 10% FCS and 1 mM sodium pyruvate. Primary cultured hepatocytes were isolated from 8-week-old male C57BL/6 mice as previously reported[37]. Isolated hepatocytes were seeded at $1 \times 10^4$ cells/well on 96-well plates and cultured in DMEM with 10% FCS for 2 h, and then in serum-free DMEM medium overnight.

Ferrostatin-1 (Fer-1, #17729) and AY9944 were purchased from Cayman Chemical (Ann Arbor, MI). Artesunate (A2191), RSL-3, and 7-DHC were purchased from Tokyo Chemical Industry (Tokyo, Japan), AdooQ Bioscience (Irvine, CA), and AdipoGen (Farmingdale, NY), respectively. Mβ-CD and other reagents were obtained from Sigma-Aldrich (St. Louis, MO) unless otherwise specified. Fer-1 and RSL-3 were dissolved in dimethyl sulfoxide (DMSO). 7-DHC was used by preparing a water-soluble inclusion complex with hydroxypropyl β-cyclodextrin and BHT-free tetrahydrofuran, both of which were purchased from Fuji Film (Tokyo, Japan)[38]. Other reagents were dissolved in PBS. In in vitro experiments, cells were treated with Fer-1, AY9944, 7-DHC, and Mβ-CD 1 h prior to RSL-3 treatment unless otherwise specified.

## Plasmids
The sgRNA targeting each gene was designed using either CRISPR direct (http://crispr.dbcls.jp) or CRISPR design tool in Benchling (https://www.benchling.com). For lentiviral vector production, the sgRNAs were subcloned into LentiCRISPRv2, which was a gift from Feng Zhang (Addgene plasmid #52961; http://n2t.net/addgene: 52961; RRID: Addgene_52961), or its derivative carrying blasticidin- or neomycin-resistant gene. LentiCRISPR vectors harboring tandem sgRNA were constructed using Gibson assembly cloning kit (New England BioLabs, Ipswich, MA). For hydrodynamic injection, gRNA and its scaffold were subcloned into pX330-U6-Chimeric_BB-CBh-hSpCas9 (Addgene plasmid #42230; RRID: Addgene_42230). Polymerase chain reaction (PCR)-generated cDNAs encoding the nuclear form of human SREBP-1a (1-460) was Gibson subcloned into CS-IV TRE CMV KT, kindly provided by Dr. H. Miyoshi (RDB12876, RIKEN BRC, Tsukuba, Japan).

## Lentiviral preparation
LentiX293T cells were co-transfected together with LentiCRISPRv2, pLP1, pLP2, and pVSVG using PEI MAX (Polysciences, Warrington, PA, USA) to prepare the lentiviral vectors. Culture media containing the lentiviral vectors were collected 3 days after transfection. The collected media were filtered with a 0.45-μm filter and ultracentrifuged at 53,600 xg using an SW55 Ti rotor (Beckman Coulter, Brea, CA, USA), and the pellets were resuspended in PBS containing 5% FCS. The lentivirus titer was measured using a Lentivirus qPCR Titer kit (Applied Biological Materials, Richmond, BC, Canada).

## Genome-wide CRISPR screening
Genome-wide CRISPR screening was performed using the Human GeCKOv2 CRISPR knockout pooled library (Addgene #1000000048). For sgRNA library transduction, $1 \times 10^7$ Huh-7 cells were infected with GeCKOv2 library virus at a target MOI of 0.3 in the presence of 8 μg/mL polybrene. Infected cells were selected with 2 μg/mL of puromycin for 3 days. The positively selected cells were expanded and treated with either 0.1 μM RSL3 or 0.03 μM RSL3 in the presence of 50 mM linoleic acid (LA) for 24 h. The surviving cells were further expanded and selected by repeated stimulation for 3 times and 5 times, separately. After selection, the genomic DNA was extracted from the surviving cells with phenol-chloroform. The sgRNA cassette was PCR-amplified and barcoded with sequencing adaptors using KOD One PCR Master Mix (TOYOBO). The PCR products were purified with a GEL/PCR purification column (FAVORGEN), quantified by Quant-iT PicoGreen dsDNA reagents and kits (Thermo Fisher), and sequenced on a Miseq sequencer (Illumina), loaded at 30% spike-in of PhiX DNA. The sequence data were processed with the MAGeCK algorithm[39].

## CRISPR/Cas9-mediated genome editing
The sgRNAs targeting DHCR7, GPX4, TM7SF2, LBR, EBP, and SC5D are listed in Supplementary Table 1. For lentiviral transduction, Huh-7 cells were incubated with lentiviral vectors for 16 h in the presence of 8 μg/mL polybrene. The transduced cells were selected by incubating them with 2 μg/mL puromycin for at least 2 days. For transduction of multiple sgRNAs, Huh-7 cells were further transduced with Lenti-CRISPR harboring the drug-resistance genes and selected by incubating them with 10 μg/mL blasticidin or 1 mg/mL G418. GPX4-ablated cells were maintained in DMEM with 10% FCS in the presence of a low dose (0.1 μM) of Fer-1.

## Cell death assay
Cytotoxicity was determined as lactate dehydrogenase (LDH) activity in cultured supernatants using a cytotoxicity detection kit (#11644793001; Roche, Mannheim, Germany), according to the manufacturer's instructions. To determine the total cellular LDH activity, cells were lysed with 2% TritonX100, and L-lactate dehydrogenase from rabbit muscle (Roche) was used as a standard. Cell death was also assessed by SYTOX Green (#S7020; Thermo Fisher Scientific), a membrane-impermeable DNA dye that enters dead cells. Nuclei were co-stained with Hoechst 33342 (No. 346-07951, Dojindo, Kumamoto, Japan). Images were captured by using confocal microscopy (FLUO-VIEW FV10i; Olympus, Tokyo, Japan). Cell viability was determined by the MTT [3-(4,5-di-methylthiazol-2-yl)-2,5-diphenyltetrazolium bromide] (Invitrogen, Waltham, MA) reduction assay. Cytotoxicity or cell viability was evaluated 24 h after RSL-3 treatment or 48 h after Fer-1 withdrawal in GPX4-ablated Huh-7 cells.

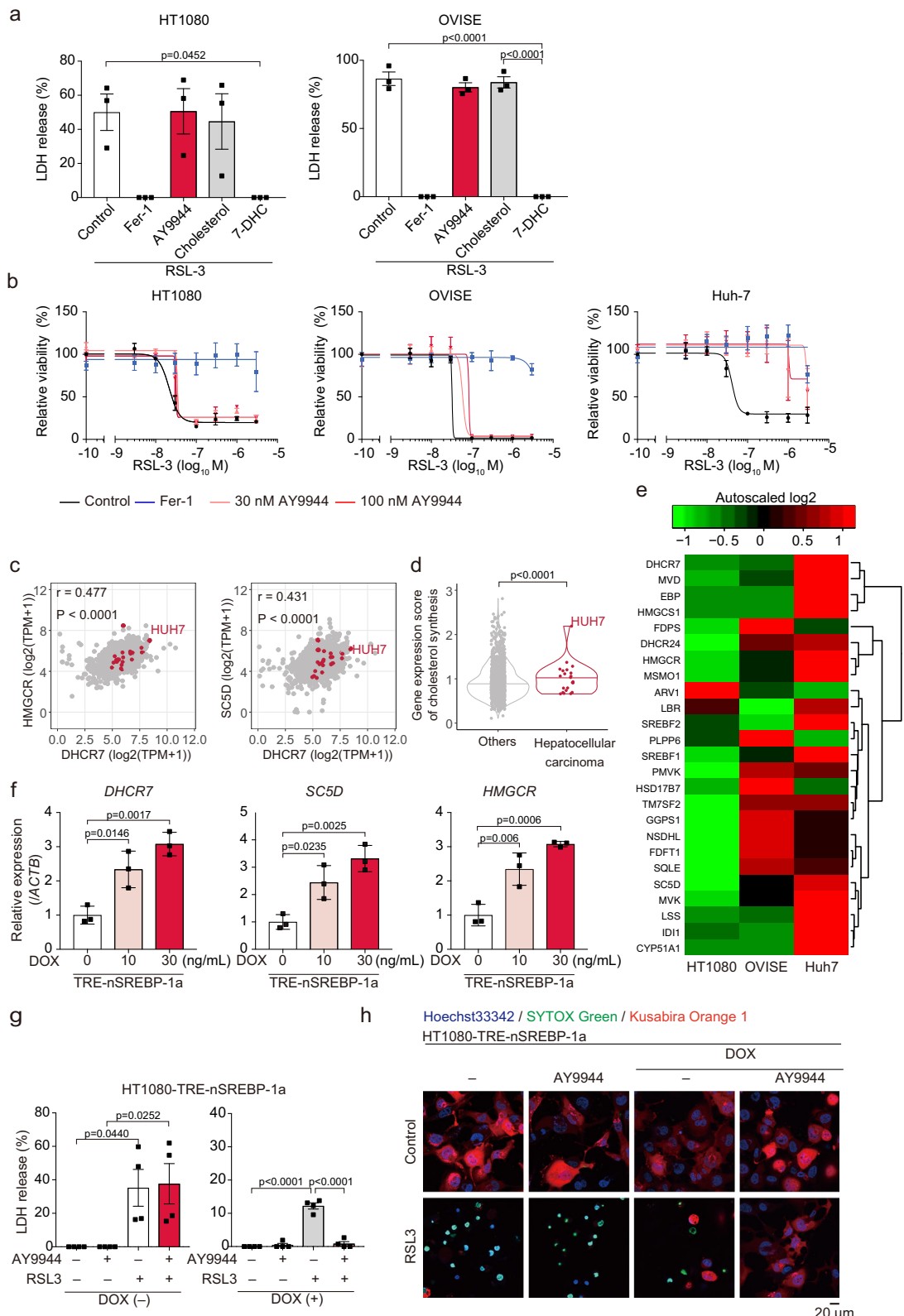

## Amplicon sequence

The frequency of indel in *DHCR7, SC5D, TM7SF2, LBR, or EBP*-ablated cells was analyzed by deep sequencing analysis. The mutated regions of each gene were PCR-amplified and barcoded with sequencing adaptors using KOD One PCR Master Mix. The primers are listed in Supplementary Table 3. The amplicons were sequenced on a Miseq sequencer (performed by Bioengineering Lab. Co., Ltd., Kanagawa,

Japan). The frequency of indel was analyzed using CRISPResso2 (version 2.2.14)[40].

## Enzymatic activities of DHCR7

Enzymatic activities of DHCR7 were determined by conversion of ergosterol to brassicasterol[41]. The detailed protocol is described in the supplementary file.

**Fig. 5 | Ferroptosis suppression by DHCR7 inhibition depends on the activity of sterol synthesis and DHCR7 expression. a** HT1080 cells and OVISE cells were treated with Fer-1 (1 μM), AY9944 (30 nM), Cholesterol-HPβCD inclusion complex, or 7-DHC-HPβCD inclusion complex for 1 h, followed by RSL-3 (0.1 μM) for 24 h. Cytotoxicity was assessed by an LDH release assay. **b** HT1080, OVISE, and Huh-7 cells were pretreated with Fer-1 (0.5 μM) or AY9944 (30 and 100 nM) for 1 h, and then treated with RSL-3 (0.1 μM) for 24 h. Cell viability was assessed by the MTT assay. **c, d** Expression of genes involved in cholesterol biosynthesis in human cancer cell lines in the CCLE dataset. **c** Scatter plot showing expression levels (log2 normalized TPM) of *DHCR7* and either *HMGCR* or *SC5D*. **d** Expression score calculated using genes listed in the Cholesterol biosynthesis/Reactome. **e** Heatmap showing the expression levels of Cholesterol biosynthesis. **f** TRE-nSREBP-1a–HT1080 cells were treated with DOX (10 ng/mL or 30 ng/mL) for 6 h. The mRNA levels of *DHCR7*, *SC5D*, and *HMGCR* were assessed by real-time RT-PCR analysis. **g** and **h** TRE-nSREBP-1a–HT1080 cells were treated with or without RSL3 for 24 h after 6 h of treatment with DOX (30 ng/mL) and AY9944 (100 nM). Cytotoxicity and cell death were assessed by **g** an LDH release assay and **h** SYTOX Green. **a, b, f**, and **g** Data are means or **h** representative of (**a, b, f**, and **h**) three or **g** four independent experiments. Statistical significance was calculated using one-way or two-way ANOVA with Tukey's post hoc test. Data are expressed as dot plots and means ± SEM.

## Assessment of lipid peroxidation

Cells were cultured overnight and labeled with 1 μM C11-BODIPY[581/591] (#D3861; Thermo Fisher Scientific) for 1 h before RSL-3 treatment. After the treatment, cells were detached and examined by flow cytometry (FACS Verse, BD Biosciences, San Jose, CA). The data were analyzed using FlowJo software (version 10; Tree Star, Inc., San Carlos, CA).

## LC-MS/MS analysis for 7-DHC and DHCEO

Intracellular 7-DHC and DHCEO levels were analyzed by liquid chromatograph-mass spectrometry (LC-MS/MS). The detailed protocol is described in the supplementary file.

## Electron spin resonance

Arachidonic acid (250 μM; AA, Sigma-Aldrich), lipoxygenase from Glycine max (soybean) (15 μg/ml; LOX, Sigma-Aldrich), and 4-hydroxy-2,2,6,6-tetramethylpiperidine-N-oxyl (50 μM; TEMPOL, Tokyo Chemical Industry, Ltd.) were mixed in 10 mM phosphate buffer (pH 7.4). As a positive control, we used liproxstatin-1 (100 μM; Lip-1, Sigma-Aldrich). ESR spectra of TEPOL were monitored using an X-band (9.45 GHz) ESR spectrometer (JES-FA100; JEOL Ltd., Tokyo, Japan) at 20 °C, microwave power at 4 mW, and an external magnetic field range of 5 mT.

## Fluorescence measurement of NBD-Pen

Arachidonic acid (500 μM; AA, Sigma-Aldrich), lipoxygenase from Glycine max (soybean) (5 μg/ml; LOX, Sigma-Aldrich) and 2,2,6-trimethyl-4-(4-nitrobenzo[1,2,5]oxadiazol-7-ylamino)-6-pentylpiperidine-1-oxyl (5 μM; NBD-Pen) were mixed in 10 mM phosphate buffer (pH 7.4). Fluorescence intensities of NBD-Pen were measured with a multimode plate reader (EnSpire; PerkinElmer Inc., Waltham, MA, USA) at excitation and emission wavelengths of 470 and 530 nm, respectively.

## LC-MS/MS analysis for lipid peroxidation products

LC-MS/MS was performed using an LCMS-8060 mass spectrometer (Shimadzu Corp., Kyoto, Japan) equipped with an electron spray ionization source. The extracted solution was measured using LC/MS/MS in the multiple reaction monitoring mode. The LC conditions were as follows: injection volume, 5 μL; autosampler temperature, 4 °C; column, InertSustain C18 column (2.1 mm × 150 mm, 3 μm particle size, GL Sciences, Tokyo, Japan); column temperature, 40 °C; mobile phase, 5 mM ammonium formate in acetonitrile:$H_2O$ = 2:1 (A) and isopropanol:methanol = 95:5 (B); flow rate, 0.4 mL/min; and gradient elution, 0–22.5 min 0–100% B, 22.5–27.5 min 100% B. The LC-MS/MS analyses were carried out using LabSolutions version 5.80 (Shimadzu Corp.). The multiple reaction monitoring (MRM) transition and parameters were listed in Supplementary Table 4.

## Cancer cell line encyclopedia (CCLE) database analysis

We analyzed the expression of genes listed in the Cholesterol biosynthesis gene set (R-HSA-191273, reactome) across multiple types of cancer cell lines using RNAseq data in the CCLE database. The log2-normalized expression of genes was visualized, and gene expression scores were calculated as a geometric mean for each gene.

## Analysis of RNA-seq data

Raw read data of liver-specific Dhcr7 knockout mice and their control mice were downloaded from sequence read archive database (PRJNA610810). The FASTQ files were aligned to GRCm38 reference genome by using HISAT2-2.2.1[42] and counted by featureCounts (subread 2.0.1)[43]. Differential gene expression analysis was performed using the DEseq2 R packages (version 1.32.0)[44]. GSEA was performed using the Broad Institute GSEA4.1.0 (http://www.gsea-msigdb.org/gsea/index.jsp) with a curated gene set. FDR-q values were computed using 1000 geneset permutations.

## Animal experiments

All experiments in this study were performed in accordance with the Jichi Medical University Guide for Laboratory Animals (Permit Nos. 17141-02 and 20107-02). C57BL/6J mice were purchased from SLC Japan (Shizuoka, Japan). Mice were housed (4/cage, RAIR HD ventilated Micro-Isolator Animal Housing Systems, Lab Products, Seaford, DE) in an environment maintained at 23 ± 2 °C with *ad libitum* access to normal chow (MFG, Oriental Yeast CO., Japan) and water under a 12-h light/dark cycle with lights on from 8:00 to 20:00. Partial hepatic ischemia was produced as previously described with some modifications[45,46]. Briefly, mice were anesthetized with isoflurane. A midline laparotomy was performed and an atraumatic clip (Fine Science Tools, Foster City, CA) was placed across the portal vein, hepatic artery, and bile duct to interrupt blood supply to the left lateral and median lobes (~70%) of the liver. After 60 min of partial hepatic ischemia, the clip was removed to initiate reperfusion. Sham control mice underwent the same protocol without vascular occlusion. Mice were sacrificed 3 h after reperfusion, and samples of blood and ischemic lobes were collected. AY9944 (25 mg/kg) or vehicle (PBS) was administered intraperitoneally 1 h prior to open laparotomy. For hydrodynamic tail-vein injection, px330 vectors encoding a pair of sgRNA targeting mDhcr7 or empty gRNA scaffolds were prepared using the EndoFree Maxi Kit (QIAGEN). Ten μg of plasmid vectors were diluted in saline (80 μL/g) and injected within 10 seconds. Seven days after injection, mice were fasted for 16 h and intraperitoneally injected with APAP (300 mg/kg). Mice were sacrificed 4 h after the injection, and samples were collected. Serum levels of AST and ALT were measured using a Fuji-DRYCHEM chemical analyzer (Fuji Film) according to the manufacturer's instructions. The paraffin-embedded tissue sections were stained with hematoxylin and eosin (HE). The severity of the liver injury was graded according to a previous report[47].

## Statistics

Data are expressed as the mean ± standard error of the mean (SEM). Differences between two groups were determined by Mann-Whitney's U test. Differences between multiple groups were determined by one-way analysis of variance (ANOVA) or two-way ANOVA combined with Tukey's post hoc test. All analyses were performed using GraphPad Prism version 7 (San Diego, CA). A *p*-value of <0.05 was considered to be statistically significant.

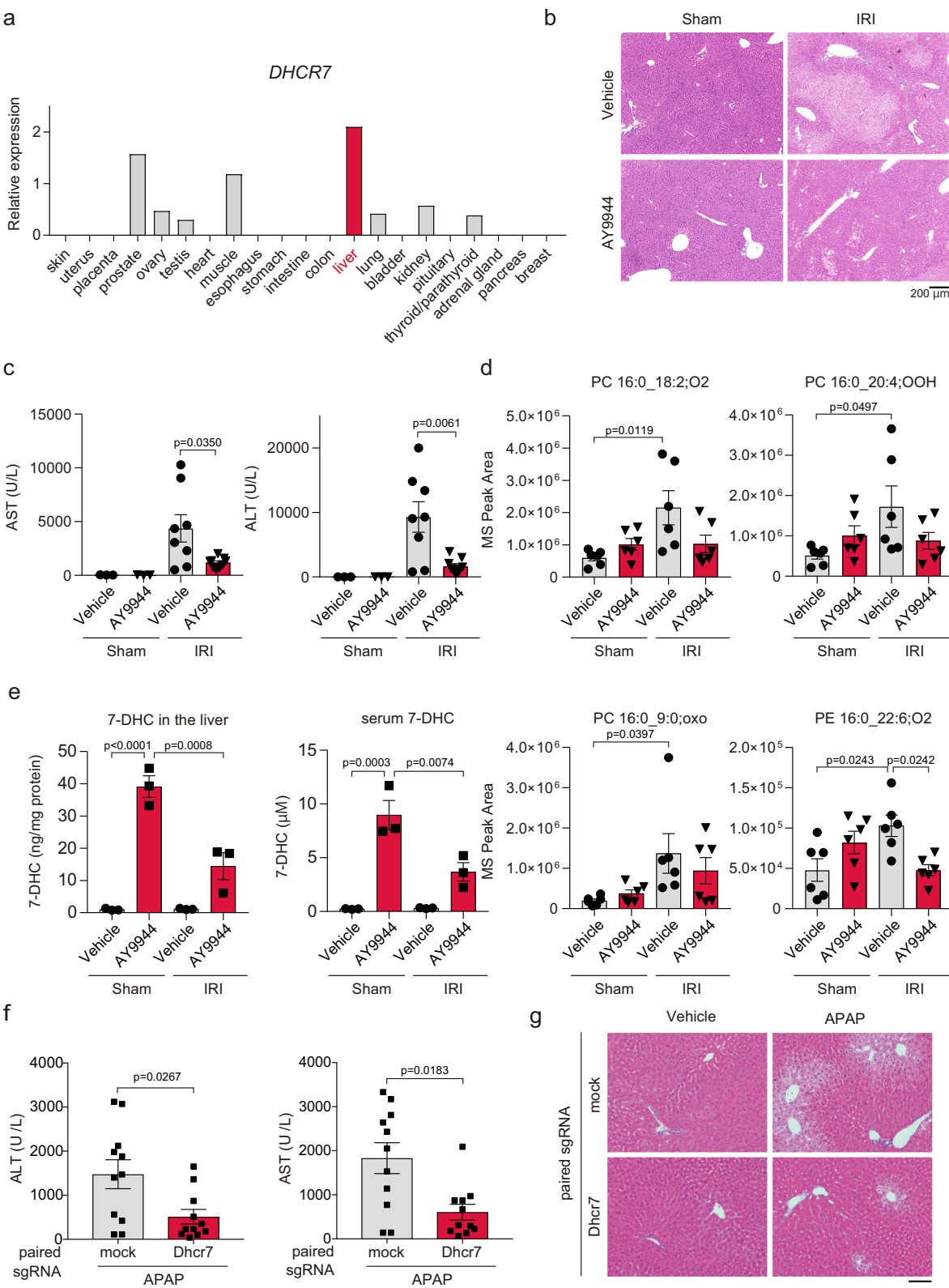

**Fig. 6 | DHCR7 inhibition prevents ferroptosis-related liver diseases. a** *DHCR7* expression in normal tissues and organs was analyzed using the RefEx database. **b–e** Liver samples and serum were obtained from hepatic ischemia-reperfusion or sham-operated C57BL/6J male mice treated with AY9944 (25 mg/kg) 1 h prior to open laparotomy. HE staining (**b**) and serum AST and ALT levels (**c**) were assessed. **d** Lipid peroxides in the liver and **e** 7-DHC in the liver and serum were assessed by LC-MS/MS. **f, g** C57BL/6J male mice were injected with 10 μg of px330 carrying a pair of Dhcr7-targeting gRNA or empty gRNA scaffold (NC) using a hydrodynamics-based procedure. PBS or acetaminophen (300 mg/kg) was intraperitoneally injected 7 days after gene transduction. Serum AST and ALT levels and HE staining were assessed. Data are collected from **c** 3–9 mice per group (*n* = 3 in vehicle/sham and AY9944/sham, *n* = 8 in vehicle/IRI, *n* = 9 in AY9944/IRI), **d** 6 mice per group, and **e** 3 mice per group. Statistical significance was calculated using two-way ANOVA with Tukey's post hoc test. **f–g** Data are from 11 mice per group and statistical significance was calculated using two-tailed Mann-Whitney's U test. Data are expressed as dot plots and/or means ± SEM.

## Reporting summary

Further information on research design is available in the Nature Portfolio Reporting Summary linked to this article.

## Data availability

The sequence data generated from genome-wide CRISPR screening in this study have been deposited in the Gene Expression Omnibus dataset under accession number: GSE228883. Source data are provided with this paper.

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

## Acknowledgements

We thank Drs. Tadashi Kasahara (Jichi Medical University), Kazumoto Murata (Jichi Medical University), Hitoshi Endo (Jichi Medical University), Yasushi Saga (Jichi Medical University), Kenji Tago (Jichi Medical University), Shinya Watanabe (Jichi Medical University), Hitoshi Shimano (University of Tsukuba), and Hidetoshi Hayashi (Nagoya City University) for their invaluable suggestions. We also thank Mika Hayashi (Jichi Medical University) for the technical assistance. This study was supported by grants from the Japan Society for the Promotion of Science (JSPS) through Grants-in-Aid for Scientific Research (20K22940 to NY; 21K08114 to MT), AMED-CREST grant (JP21gm0910013 to KY) the Kurata Grants from the Hitachi Global Foundation (NY), the Japan Society for Organ Preservation and Biology (NY), and the Smoking Research Foundation in Japan (MT), and Takeda Science Foundation (TK).

## Author contributions

N.Y., T.K., and M.T. conceived the study and wrote the manuscript. N.Y. and T.K. performed in vitro and in vivo experiments. T.N., T.K, C.B., and T.M., assisted cell experiments. J.I., D.Y., K.Morimoto, Y.S. Y.J., N.K., K.Y., N.Yahagi, and S.I. performed experiments using mass spectrometry and/or ESR. K.W. helped genome editing in mice.

## Funding

## Competing interests

The authors declare no competing interests.

## Additional information

¹Division of Inflammation Research, Center for Molecular Medicine, Jichi Medical University, Shimotsuke, Tochigi, Japan. ²Division of Gastroenterological, General and Transplant Surgery, Department of Surgery, Jichi Medical University, Shimotsuke, Tochigi, Japan. ³Institute of Metabolism and Cell Death, Molecular Target and Therapeutics Center, Helmholtz Munich, Neuherberg, Bavaria, Germany. ⁴Laboratory of Food Function Analysis, Graduate School of Agricultural Science, Tohoku University, Sendai, Miyagi, Japan. ⁵Division of Endocrinology and Metabolism, Department of Medicine, Jichi Medical University, Shimotsuke, Tochigi, Japan. ⁶Department of Molecular Pathobiology, Faculty of Pharmaceutical Sciences, Kyushu University, Fukuoka, Fukuoka, Japan. ⁷Division of Human Genetics, Center for Molecular Medicine, Jichi Medical University, Shimotsuke, Tochigi, Japan. ⁸Division of Gastroenterology, Department of Medicine, Jichi Medical University, Shimotsuke, Tochigi, Japan. ⁹These authors contributed equally: Naoya Yamada, Tadayoshi Karasawa. ✉e-mail: naoya.yamada@helmholtz-munich.de; tdys.karasawa@jichi.ac.jp; masafumi2@jichi.ac.jp

