## [Peer Review File · Nature Communications]

Inhibition of 7-dehydrocholesterol reductase prevents hepatic ferroptosis under an active state of sterol synthesisREVIEWER COMMENTS

Reviewer #2 (Remarks to the Author):

This reviewer previously raised concerns about the incomplete knockout of DHCR7 and other enzymes involved in cholesterol synthesis in Huh7 cells. In this revision, the authors explained that they performed most of their experiments in the polyclonal DHCR7-knockout cells in which some residual DHCR7 can still be found. To address this critic, they generated monoclonal DHCR7-knockout cells and showed that they were also resistant to ferroptosis. If true, this observation suggests that the residual amount of DHCR7 in the polyclonal knockout cells is not sufficient to restore the ferroptotic sensitivity. The problem remaining on this issue is that they only showed a sequence trace file to demonstrate the nature of the mutation in the monoclonal knockout cells. An immunoblot analysis showing that the DHCR7 protein is completely absent in the monoclonal knockout cells, which is the standard practice in the field, is necessary to demonstrate the complete knockout of the gene in the cells.

In contrast to DHCR7 knockout, the authors did not address my previous concerns regarding the incomplete knockout of other enzymes involved in cholesterol synthesis to demonstrate the specificity of AY9944. No western blot results were shown to demonstrate that the targeted proteins were indeed knocked out from the cells. Without a clear demonstration of complete knockout of these genes, the specificity of AY9944 remains unresolved.

Manuscript number: NCOMMS-23-47864-T

Responses to Comments by Reviewer #2

We appreciate very much your helpful comments for the further improvement of our manuscript. According to the comments, we performed additional experiments and carefully made the necessary amendments (changes are shown in red in the revised manuscript).

Comment #1. “This reviewer previously raised concerns about the incomplete knockout of DHCR7 and other enzymes involved in cholesterol synthesis in Huh7 cells. In this revision, the authors explained that they performed most of their experiments in the polyclonal DHCR7-knockout cells in which some residual DHCR7 can still be found. To address this critic, they generated monoclonal DHCR7-knockout cells and showed that they were also resistant to ferroptosis. If true, this observation suggests that the residual amount of DHCR7 in the polyclonal knockout cells is not sufficient to restore the ferroptotic sensitivity. The problem remaining on this issue is that they only showed a sequence trace file to demonstrate the nature of the mutation in the monoclonal knockout cells. An immunoblot analysis showing that the DHCR7 protein is completely absent in the monoclonal knockout cells, which is the standard practice in the field, is necessary to demonstrate the complete knockout of the gene in the cells.”

Response:

Thank you for your comment. As the reviewer pointed out, a Western blot showing the absence of DHCR7 protein in knockout (KO) cells was lacking. We now obtained an excellent DHCR7 antibody and performed Western blot analysis in monoclonal *Dhcr7* KO Pfa1 cells and polyclonal *DHCR7* KO Huh7 cells. We confirmed that DHCR7 expression was completely abolished in *Dhcr7* KO Pfa1 cells (Supplementary Fig. 3B). Furthermore, DHCR7 expression in the polyclonal *DHCR7* KO cells was almost abrogated (Supplementary Fig. 1F). To further validate whether DHCR7 is defective in the polyclonal *DHCR7* KO cells, we assessed DHCR7 enzymatic activities by evaluating the conversion of ergosterol to brassicasterol. The enzymatic activities of DHCR7 in the polyclonal *DHCR7* KO cells were almost completely abrogated (Supplementary Fig. 1D and E). These results verify the validity of our experiments using

DHCR7 KO cells.

2. “The authors did not address my previous concerns regarding the incomplete knockout of other enzymes involved in cholesterol synthesis to demonstrate the specificity of AY9944. No western blot results were shown to demonstrate that the targeted proteins were indeed knocked out from the cells. Without a clear demonstration of complete knockout of these genes, the specificity of AY9944 remains unresolved.”

Response:

We appreciate this important comment. To confirm the specificity of AY9944, we previously developed cells lacking *TM7SF2*, *LBR*, and *EBP* which can be inhibited by AY9944 as off-targets (Supplementary Fig. 8A). According to the reviewer’s suggestion, we confirmed that *LBR* and *EBP* were indeed absent in the KO cells (Supplementary Fig. 9B). Unfortunately, however, the antibody against *TM7SF2* did not work and therefore we were not able to assess the gene disruption at protein levels. Instead, we analyzed the substrates of these enzymes. A previous report has suggested that AY9944 promotes the accumulation of 14-dehydro zymostenol (14DZyme, a substrate for *DHCR14* which is encoded by *TM7SF2* and *LBR*) and zymostenol (Zyme, a substrate for *EBP*) at a high dosage (ACS Pharmacol Transl Sci. 2022;5:1086). Therefore, we evaluated the accumulation of 14DZyme and Zyme in AY9944-treated Huh-7 cells. Consistent with this report, high dosages of AY9944 (300–1000 nM) induced the accumulation of 14DZyme. However, low dosages of AY9944 (10–100 nM, used in our experiments) did not exhibit off-target inhibition (Supplementary Fig. 8B and C). Taken together, these results indicate that the ferroptosis-inhibitory effect of AY9944 in our study is exerted by the inhibition of *DHCR7*.

We reiterate our appreciation of your observations and hope that our revision will meet your approval for publication in *Nature Communications*.

REVIEWERS' COMMENTS

Reviewer #2 (Remarks to the Author):

This revision has addressed all my concerns and the manuscript is ready for publication

Manuscript number: NCOMMS-23-47864-T (Yamada et al.)

Responses to Comments by Reviewer #2

We appreciate very much your helpful comments for our manuscript.

We reiterate our appreciation of your observations and hope that our revision will meet your approval for publication in *Nature Communications*.